# A Comprehensive Chemical and Nutritional Analysis of New Zealand Yacon Concentrate

**DOI:** 10.3390/foods12010074

**Published:** 2022-12-23

**Authors:** Keegan Chessum, Tony Chen, Rothman Kam, Mary Yan

**Affiliations:** 1Department of Food Science and Microbiology, Auckland University of Technology, Auckland 1010, New Zealand; 2Healthcare and Social Practice, TePukenga Unitec, Auckland 0612, New Zealand

**Keywords:** yacon concentrate, fructooligosaccharides, mineral profile, free amino acids, phenolic profile, organic acids, antioxidant activity, glycaemic index

## Abstract

Global interest in yacon (*Smallanthus sonchifolius*) is growing due to its potential as a functional food, attributable to its unique profile of bioactives and high fructooligosaccharide (FOS) content, which vary between cultivars. Our objective was to conduct a comprehensive chemical and nutritional analysis of New Zealand yacon concentrate (NZYC)—a sweet syrup derived from the roots of cultivar ‘New Zealand’, which was first grown in the 1980s. The major minerals in NZYC were potassium, phosphorus, and calcium. The FOS content ranged from 17.6 to 52.7 g/100g. Total phenolic content ranged from 565 to 785 mg gallic acid equivalents per 100 g; chlorogenic acid and caffeic acid were the major phenolic compounds. The major amino acids were L-arginine, L-glutamic acid, L-proline, L-aspartic acid, and asparagine. The major organic acids were citric, malic, quinic, and fumaric acids. Antioxidant activity ranged from 1084.14 to 3085.78 mg Trolox equivalents per 100 g depending on the assay used. The glycaemic index (GI) value was 40 ± 0.22, classifying it as a low-GI food. These results support the classification of NZYC as a nutraceutical food product for future diet therapy applications.

## 1. Introduction

Yacon (*Smallanthus sonchifolius*) is an herbaceous perennial plant which forms tuberous roots [1,2]. It is native to the lower latitudes of South America (0–25 °S), and grows in subtropical and warm temperate environments at altitudes between 600 and 3500 m [3]. Optimum growth occurs between 18 and 25 °C, while temperatures below 10 to 12 °C can be detrimental to growth [3].

In the early 1980s, yacon was imported into New Zealand as a novel vegetable [1,3]. An unpublished study carried out in New Zealand planted yacon in four sites at latitudes from 37 to 45 °S. The two South Island sites (Lincoln and Mosgiel) had very low yields compared to the North Island sites of Pukekohe and Hamilton. This can be attributed to the climatic differences between the North and South Islands, wherein the higher temperatures and longer season in the North Island matches more closely to the Andean conditions [3]. One study on the chemical composition of seven different yacon cultivars from around the world showed that yacon tubers of the ‘New Zealand’ cultivar had the lowest sum of glucose, fructose, and sucrose, and the lowest total phenolic content, total flavonoid content, 2,2-diphenyl-1-picrylhydrazyl (DPPH) radical scavenging activity, and ferric reducing antioxidant power (FRAP); however, ‘New Zealand’ yacon tubers did have the highest 2,20-azino-bis(3-ethylbenzothiazoline-6-sulfonic acid) (ABTS) radical scavenging activity [4].

In recent years, there has been growing global interest in yacon due to its potential as a functional food, which can be related to its unique profile of bioactive and prebiotic compounds including antioxidants and phenolic compounds [1,2,3]. Of especial interest is the concentration of fructooligosaccharides (FOS) in the yacon tubers, comprising between 40 and 70% of dry matter by weight, which is higher than that of any other plant matter [1,2].

Fructooligosaccharides are short-chain oligosaccharide fructans with a degree of polymerisation (DP) lower than 9, classed as non-digestible prebiotic carbohydrates. They are non-digestible as amylase enzymes are unable to hydrolyse β-(2,1) bonds; similarly, salivary and internal digestive enzymes are unable to hydrolyse FOS [2,5]. Thus, FOS have a low calorific value for humans, and pass undigested through to the colon. In the colon they are fermented by anaerobic bacteria, forming short-chain fatty acids (SCFA). Animal studies suggest that the formation of SCFA can increase local immune response and reduce colon pH, suppressing inflammation and reducing the risk of colorectal cancer [2]. Furthermore, consumption of FOS supports the growth of beneficial gut bacteria (particularly those of the genus *Bifidus* and *Lactobacillus*), which may aid in the easing of constipation and reduction of lipid and glucose in the blood [2,5]. Conversely, in the few clinical studies carried out on yacon, only bloating and flatulence have been reported as adverse effects attributable to FOS content, and only in levels of daily intake above 0.14 g FOS/kg body weight; the only significant concern related to consumption of yacon is one reported case of anaphylaxis [5]. Yacon roots may also contain inulin; like FOS, inulins are fructan-type oligosaccharides—however, they can have DP of up to 60. Inulins are known to have similar health-promoting effects to those of FOS [2].

One product that has recently arisen in the market as a result of growing interest in yacon is yacon syrup or yacon concentrate, which is produced from juice extracted from the roots of the yacon plant [1,3]. The roots are washed and disinfected, peeled, and mechanically juiced [6]. The juice is filtered and then evaporated and concentrated to a pre-syrup, which is then filtered again and concentrated to the final syrup [6]. The syrup then undergoes a final filtration before packaging [6]. Yacon concentrate is associated with physical and sensorial characteristics similar to that of honey or sugar cane syrup [5]. Unlike other sweet syrups, yacon concentrate is hypocaloric owing to its high FOS content; furthermore, dietary supplementation of yacon concentrate has been shown to enhance satiety sensation, aiding in weight loss [5]. Thus, yacon concentrate may be considered as a nutraceutical food, and has potential for development of novel food products and new diet therapy applications [2,5].

In terms of proximate composition, carbohydrates make up 65–70% of yacon concentrate by weight, while water accounts for around 25%, protein for 1–2%, and fat for around 0.1% [6]. The only significant micronutrient in yacon concentrate is potassium, which accounts for around 1% of the concentrate total weight [6]. The carbohydrate profile of yacon concentrate can vary widely between varieties of cultivars; for instance, yacon concentrate produced from cultivar CLLUNC118 contains 10.9% FOS, 15.5% free glucose, 25.4% free fructose, and 12.2% free sucrose, while yacon concentrate produced from cultivar AMM5163 contains 47.6% FOS, 2.6% free glucose, 7.9% free fructose, and 20.0% free sucrose [6].

Phenolic compounds are secondary metabolites in plants which are of interest primarily due to their antioxidative properties, as well as their anti-inflammatory and anti-carcinogenic properties [2,4]. Yacon tubers, from which yacon concentrate is derived, contain a notably high level of phenolic compounds—approximately 200 mg per 100 g fresh weight [6]. Chlorogenic acid has been identified as a major phenolic compound present in yacon tubers [6,7]; other phenolic compounds identified include quercetin, ferulic acid, quinic acid, 3,5-dicaffeoylquinic acid, caffeic acid, and three ester derivatives of caffeic acid [1,6,8]. Amino acids, similar to phenolic compounds, are of interest primarily due to their antioxidative properties. The major amino acid identified in yacon concentrate in the literature is L-tryptophan; glutamine, arginine, alanine, threonine, and valine have also been identified as present in yacon concentrate [7,9]. The organic acids formic acid, fumaric acid, citric acid, and malic acid have also been identified as being present in yacon concentrate [9].

Glycaemic index (GI) is defined as the incremental blood glucose area following ingestion of a food of interest, expressed as a percentage of the corresponding area following ingestion of the equivalent amount of carbohydrate from a standard reference product [10]. Low GI foods are those for which the breakdown of carbohydrates into glucose and subsequent absorption into the blood takes relatively longer and have GI values of 55 or less. Low GI diets have been shown to improve overall blood glucose and lipid control for normal and diabetic individuals, and can aid in weight loss, reduction of circulating triglycerides, and improvement of blood pressure [11,12].

The aim of this present study is to produce a comprehensive chemical and nutritional profile of yacon concentrate derived from the ‘New Zealand’ yacon cultivar, as there is currently a paucity of data on ‘New Zealand’ yacon concentrate in the literature. This will confirm the potential of this yacon concentrate as a nutraceutical food product and as an ingredient to add value to other products such as functional drinks.

## 2. Materials and Methods

The yacon concentrate used in this study was produced from yacon cultivar ‘New Zealand’. Three different production batches were sourced from Yacon New Zealand Ltd. Yacon crops are grown in the North Island of New Zealand and typically harvest in the mid-year. After harvest, the yacon roots are graded, milled, pressed to juice, and concentrated to 80° Brix.

All reagents used in this study are ≥99% in purity, unless otherwise stated. Refer to Appendix A for list of chemicals and reagents used for this study.

Protein content of NZYC was determined by the Kjeldahl method [13]. Digestion was carried out using a DK 20 heating digester (Velp Scientifica, Via Stazione, Italy) equipped with a JP reticulating water aspirator (Velp Scientifica, Italy), and the distillation setup consisted of a butane gas canister (Gasmate, Hamilton, New Zealand), Kjeldahl flask, Kjeldahl trap, and Graham condenser.

Moisture content was determined by refractive index using a J57 automatic refractometer (Rudolph Research, Hackettstown, NJ, USA) [14]. Ash content was determined by mass difference using a Perfect Fire HDTP-56-55 furnace (Canadian Instrumentation Company, Calgary, AB, Canada) [14]. Carbohydrate content was determined by mass difference once protein, moisture, and ash content were determined.

The mineral profile of yacon concentrate was determined by microwave plasma atomic emission spectrometry (MP-AES) and the methodology is described elsewhere [14].

The concentrations of fructose, glucose, and sucrose present in yacon concentrate were determined by high performance liquid chromatography coupled with and electronic light scattering detector (HPLC/ELSD) [14], using an initial sample weight of 0.20 g. HPLC/ELSD instrumentation and analysis parameters were described in the literature [14].

The concentrations of fructooligosaccharides and inulins (degree of polymerisation [DP] 3–13) was determined by HPLC-ELSD. A mixed standard of 1-kestose (DP3) and nystose (DP4) was prepared at a series of concentrations from 1.5 to 0.03125 ppm in ultra-pure water (UPW). Approximately 0.3 g yacon concentrate was dissolved in 10 mL of UPW. 3 mL of sample was washed three times with 3 mL of chloroform to remove all non-polar organic compounds and the rest of the procedure is described elsewhere [14]. Except, that the pump flow rate was 1.0 mL/min, analysis time of 62 min and the injection volume of 20 µL were used.

The total phenolic content of NZYC was determined by the Folin–Ciocalteu (FC) [14], with minor modifications. 500 µL of FC phenol reagent was added to 20 µL of prepared yacon concentrate sample and diluted with 980 µL UPW with all other reaction steps the same [14].

The phenolic profile of NZYC was determined by liquid chromatography-mass spectrometry (LC-MS) [14]. For the standards, a 100 mg/L stock solution of a mixture of phenolic compounds (chlorogenic acid, gallic acid, catechin, epicatechin, caffeic acid, p-coumaric acid, ferulic acid, ellagic acid, rutin, kaempferol, kaempferol rutinoside, isorhamnetin, and myricetin) was prepared in methanol, and diluted with methanol to yield standards at concentrations ranging from 0 to 20 mg/L. Stock and standard solutions were stored in the dark at 4 °C. The LC-MS instrumentation, MS ionization source conditions, and LC-MS conditions were as described in the literature [14]. The gradient program was as follows: The initial percentage of mobile phase B (B%) was 5%, and held for 0.5 min. B% was then raised to 15% over 1.5 min. B% was then raised to 20% over 7 min; then to 50% over two minutes and held at 50% for a further two minutes. B% was then raised to 80% over one minute and held at 80% for a further two minutes. B% was then lowered to 5% over the course of one minute and held until the run time of 28 min was complete.

For antioxidant analysis, 6-hydroxy-2,5,7,8-tetramethylchroman-2-carboxylic acid (Trolox) standards were prepared within the concentration range of 5 to 160 mg/L in 75% ethanol. Yacon concentrate sample was prepared by making 1.0 g of yacon concentrate up to 10 mL with UPW. The antioxidant activity of yacon concentrate was determined by the FRAP assay [14]. The antioxidant activity of yacon concentrate was determined by the CUPRAC assay [14].

The antioxidant activity of yacon concentrate was determined by the DPPH assay [15,16]. A 0.1 M sodium acetate buffer was prepared by dissolving sodium acetate in UPW. A buffered ethanol solution was then prepared by mixing 0.1 M sodium acetate buffer with ethanol in a 2:3 ratio by volume. A 0.3 mM DPPH solution was prepared by dissolving DPPH in methanol; this solution was stored at 4 °C in the dark as it was light and temperature sensitive. A 100 mg/L Trolox stock solution was prepared by dissolving Trolox in buffered ethanol, from which a series of standards were produced at concentrations from 2 to 40 mg/L. Yacon concentrate stock solutions were prepared by dissolving one gram of yacon concentrate in 25 mL buffered ethanol; sample solutions were prepared by further dilution with buffered ethanol, ranging from ten-fold to two-fold dilutions. To perform the assay, 0.20 mL of diluted sample, standard, or buffered ethanol (control) was mixed with 2 mL of DPPH, then 2 mL of methanol was added. For the blank, 0.20 mL of buffered ethanol was mixed with 4 mL of methanol. The reaction mixtures were kept in the dark at room temperature for 30 min, before absorbance was read at 517 nm against the blank.

DPPH inhibition was calculated according to Equation (1), where *A_c_* is the absorbance of the control and *A_s_* is the absorbance of the sample or standard. Maximum inhibition percentage was determined as 50% inhibition; above 50%, the absorbance reading is meaningless.
(1)$$Inhibition\;of\;DPPH\;\left(\%\right)=\left(\frac{A_{c}-A_{s}}{A_{s}}\right)*100$$

The amino acid profile of yacon concentrate was determined by the AccQTag (6-aminoquimolyl-N-hydroxysuccinimidyl carbamate) derivitisation method and analysed using liquid chromatography-mass spectrometry (LC-MS), using an initial sample weight of 0.1 g [14]. LC-MS instrumentation, MS ionisation source conditions, and LC-MS conditions were as described in the literature [14], except the column used was a Poroshell 120 EC-C18 (2.1 × 150 mm, 2.7µm) (Agilent, Santa Clara, CA, USA), and the LC-MS flow rate was 0.3 mL/min. The gradient program was as follows: The initial percentage of mobile phase B (B%) was 5%, and held for 1 min. B% was then raised to 10% over four minutes, then to 15% over five minutes. B% was then raised to 45% over five minutes, then to 80% over two minutes and held at 80% for a further two minutes. B% was then lowered to 5% over one minute and held at 5% until the run time of 29 min was complete.

A quantitative organic acid profile of yacon concentrate was produced using LC-MS. A mixed standard (citric, succinic, malic, maleic, malonic, fumaric, tartaric, pyruvic, quinic, and salicylic acids) was prepared at 500 ppm. Via serial dilution, a series of mixed standards was produced at concentrations from 10 to 0.3125 ppm. Yacon concentrate (approximately 1.0 to 1.5 g) was dissolved in 10 mL of UPW, and diluted down a hundredfold in UPW. Samples were then centrifuged at 4466× *g* for 5 min, and the supernatant and mixed standards were used for analysis by LC-MS. The LC-MS instrumentation and MS ionisation source conditions used were the same as for determination of amino acid profile, except the column used was a Kinetex Evo C18 (2.1 × 150 mm, 1.7 µm) (Phenomenex, USA) and the nebulizer pressure was 40 psi. The negative and positive ionisation modes were performed with multiple reaction monitoring (MRM) for quantitative analysis. The LC-MS conditions were as follows: Mobile phase A was 0.1% formic acid in UPW. Mobile phase B was 0.1% formic acid in acetonitrile. The flow rate was 0.2 mL/min, and the column temperature was 40 °C. The gradient program was as follows: The initial percentage of mobile phase B (B%) was 5%, and held for 0.5 min. B% was then raised to 20% over two and a half minutes, then to 50% over two minutes and held at 50% for a further four minutes. B% was then lowered to 5% over three minutes and held at 5% until the run time of 25 min was complete.

To evaluate its health-related properties, NZYC was tested for its glycaemic index (GI). The test was conducted at the Sydney University’s Glycaemic Index Research service. GI was measured in vivo using the international standard method ISO 26642:2010(E) [17] (*n* = 10). A group of 10 healthy, non-smoking people were recruited for the test. A weighted potion of the NZFOS+ sample containing 25 g of available carbohydrate or a drink containing 25 g of glucose was consumed within 10 min on two separate occasions. Participants capillary blood samples were analysed for glucose at 0 (baseline), 15, 30, 45, 60, 90, 120 min after the start of the ingestion of a test food. GI was calculated as the area under the response curve and above the baseline [17]. The GI test was reviewed and approved by the Human Research Ethics Committee of the University of Sydney (2017/801). Informed consent was obtained from all participants involved in the test.

Statistical analysis was carried out using R Studio version 1.1.463. All analyses (apart from mineral profile) were done in triplicates from different batches of yacon concentrate; only one batch of yacon concentrate was used for mineral profile. One-way analysis of variance (ANOVA) and post hoc analysis (Tukey’s honestly significant difference test) were carried out using R Studio, and results were reported as mean value ± standard deviation on wet weight basis.

## 3. Results and Discussion

### 3.1. Proximate Analysis, Sugar Profile, and Glycaemic Index

The results from the proximate analysis and determination of sugar profile of NZYC are presented in Table 1. NZYC is mainly carbohydrate in nature, with content ranging from 79.454 ± 0.387 to 80.671 ± 0.664 g/100g. Fructose was the major sugar quantified, comprising between 27.783 ± 2.830 g/100g and 36.463 ± 2.757 g/100g. In batches 1 and 3 of yacon concentrate, the concentrations of glucose and sucrose were approximately double that of nystose; however, in batch 2, the concentration of nystose was higher than that of glucose and sucrose. This corresponds with batch 2 having significantly lower fructose concentration than batches 1 and 3. Additionally, the HPLC-ELSD profile for batch 2 contained more peaks than either batch 1 or batch 3. As fructooligosaccharides and inulins are fructans (chains of fructose joined together with a terminal glucose molecule), the observed inverse relationship between fructose and FOS/inulin concentration is not unsurprising. The chromatographic peaks DP5-13 were assigned based on the generally accepted assumption that the retention time of structurally similar carbohydrates will increase as DP increases [18]. However, it should be noted that commercial standard/reference compounds with DP higher than four were not obtained, and so quantification based on the standard curve developed for nystose (DP4) is only semi-quantitative for FOS with DP5+.

The sum of FOS in the yacon concentrate samples was as follows: 24.225 ± 0.809 g/100g (batch 1), 52.276 ± 0.808 g/100g (batch 2), and 17.625 ± 0.325 g/100g (batch 3). In the literature, the FOS content of yacon syrup ranges from 10.9 to 47.6% depending on a number of factors, including cultivar, physiological conditions such as plant age, harvest and post-harvest conditions, and others [9]. The results obtained in the present study fall either within this range or just above, which may be due to any of the aforementioned factors, or the semi-quantitative method used for FOS with DP5+. It can be reasonably determined that the semi-quantitative method used for higher DP sugars leads to an over-estimation of FOS content, as the total sum of sugars quantified in batch 2 is 97.669 g/100g, much higher than the total carbohydrate content of 79.454 ± 0.387 g/100g. The total sum of sugars in batches 1 and 3 are more reasonable (78.932 and 71.173 g/100g, respectively). Glycaemic index is a measure of the effect of consumption of carbohydrate foods on blood glucose level. The results from the Sydney University’s Glycaemic Index Research service revealed that yacon concentrate has a low glycaemic index (GI = 40 ± 0.22). Although other natural sweeteners such as maple syrup, coconut sugar and molasses can be classified as low-GI foods, they have higher GI values, typically around 54–55; honey is classified as a medium-GI food (GI between 56–69). This supports the classification of yacon concentrate as a nutraceutical food and its potential for future diet therapy applications.Moisture content ranged from 9.648 ± 0.320 to 11.943 ± 0.046 g/100g, whilethere were relatively low levels of protein (4.744 ± 0.650 to 6.634 ± 0.158 g/100g) and ash (2.642 ± 0.130 to 3.364 ± 0.410 g/100g). The moisture content of NZYC is much lower than that of 25% by weight as reported in the literature [6]; however, this could be due to differences in processing methodology, as the yacon concentrate in the literature was developed with the aim of achieving a final product with 73° Brix, while the yacon concentrate used in the present study claims to have 80° Brix. This claim is consistent with the determined carbohydrate content of approximately 80 g/100 g. Conversely, the determined protein content was two to three times higher than the literature value of 1–2% [6]. This difference may be attributable to a number of factors, such as differences in processing methodology, variety of yacon used, harvest time, or soil conditions when cultivating yacon or climatic conditions when nurturing the crop.

### 3.2. Mineral Profile

Fourteen minerals were quantified in yacon concentrate by microwave plasma atomic emission spectrometry (MP-AES), and the results are presented in Table 2. Potassium was the most abundant mineral (658.366 ± 5.927 mg/100 g), followed by phosphorus (93.121 ± 0.500), calcium (44.225 ± 4.478), and magnesium (36.628 ± 0.907). Nickel, zinc, copper, lead, and titanium were detected at concentrations lower than 1 mg/100 g, while antimony, boron, and cobalt were not detected. The mineral profile of NZYC is consistent with the literature, in that potassium is the only major micronutrient identified by [6], although the potassium content identified in the present study is slightly lower than 1% by weight. The mineral profile of yacon concentrate obtained in the present study is similar to the mineral profile of Brazilian yacon syrup [9]. Potassium was the major mineral (691.00 ± 33.96 mg/100 g), followed by phosphorus (162.00 ± 2.65), magnesium (45.67 ± 10.12), sulfur (42.00 ± 2.65, not quantified in the present study), calcium (40.67 ± 3.79) and sodium (17.00 ± 1.73) [9]. Copper, iron, zinc, and manganese were also quantified at relatively low concentrations [9].

### 3.3. Phenolic Profile

The phenolic profile of NZYC is presented below in Table 3, and the results from the Folin-Ciocalteau assay are presented in Table 4. Chlorogenic acid was the major phenolic compound identified (6.187 ± 0.221 to 10.361 ± 1.398 mg/100 g), with caffeic acid the only other compound quantified at concentration higher than 1 mg/100 g. p-Coumaric acid, ferulic acid, kaempferol, and isorhamnetin were all quantified at concentrations lower than 1 mg/100 g. The total phenolic content of NZYC ranged from 564.97 ± 9.28 mg GAE/100 g to 785.11 ± 43.14 mg GAE/100 g.

The phenolic profile of NZYC obtained in the present study is in agreement with the literature, in that chlorogenic acid is the major phenolic compound identified in yacon concentrate, with caffeic acid and ferulic acid also being identified as present [1,2,6,8]. Various studies on the benefits of chlorogenic acid have suggested that chlorogenic acid could attenuate cognitive decline and reduce the risk of neurodegeneration, and has potential to protect against oxidative damage, positively affect glucose tolerance, and decrease blood pressure [19]. Caffeic acid has been suggested to have anti-inflammatory, anticancer, antiviral, anti-depressive and antihyperglycemic effects [20], while ferulic acid has been shown to exert antimicrobial, anti-inflammatory, antidiabetic, and anticancer effects; studies suggest it may also be used to moderate blood pressure and provide neuroprotective effects [21].

However, the total phenolic content (TPC) of NZYC determined in the present study is surprising. The discussed literature indicated that yacon tubers of the ‘New Zealand’ variety had lower TPC than other yacon tubers, yet the TPC of NZYC had higher TPC than other yacon concentrates reported in the literature. The TPC of yacon syrup obtained from a local market in Brazil was 120.225 ± 3.005 mg GAE/100 g [22], yacon syrup produced from tubers obtained in Wosobo, Indonesia had TPC ranging from 90.765 to 105.056 mg GAE/100 g [23]. This could be attributed to differences in processing parameters—for instance, the Brazilian yacon syrup was concentrated to 71° Brix [22], while the yacon concentrate used in the present study is concentrated to 80° Brix—or various other factors such as soil and climatic conditions, and harvest and post-harvest conditions.

### 3.4. Antioxidant Activity

The antioxidant activity of NZYC was determined by three different methods; the FRAP, CUPRAC, and DPPH assays, and the results are presented below in Table 5. Each assay gave different results for the antioxidant activity of NZYC. These differences arise from the fact that each assay is based on a different chemical reaction; antioxidant compounds exert different activity, and co-existing compounds interfere differently, for each reaction [24]. According to the FRAP assay, the antioxidant activity of NZYC ranges from 1084.14 ± 18.63 to 1442.54 ± 259.11 mg TE/100 g. According to the CUPRAC assay, antioxidant activity ranges from 2735.31 ± 235.47 to 3085.78 ± 637.45 mg TE/100 g. According to the DPPH assay, antioxidant activity ranges from 1536.08 ± 64.34 to 1551.38 ± 334.70 mg TE/100 g.

New Zealand Manuka honey, considered to be the “gold standard” amongst honeys in terms of antioxidant activity, has been found in the literature to have antioxidant activity ranging from 31.74 ± 1.86 to 519.76 ± 3.56 mg TE/100 g according to the FRAP assay [25,26,27]. Goji berries, another nutraceutical food of interest, have been shown to have antioxidant activity ranging from 133.26 ± 5.28 to 486.54 ± 4.58 mg TE/100 g according to the FRAP assay [28]. In terms of the FRAP assay, NZYC has antioxidant activity two to three times higher than other foods that are of interest due to their antioxidant activities. The antioxidant activity of New Zealand Manuka honey as determined by the CUPRAC assay is 52.6 mg TE/100 g [29], while goji berries have been shown to have antioxidant activity ranging from 154.35 ± 0.83 to 264.66 ± 1.78 mg TE/100 g according to the CUPRAC assay [28]. In terms of the CUPRAC assay, NZYC has antioxidant activity at least ten times greater than other foods of interest due to their antioxidant activities. Studies on New Zealand Manuka honey found antioxidant activity to range from 11.17 ± 0.80 to 28.41 ± 1.72 mg TE/100 g according to the DPPH assay [26,30], while goji berries have been shown to have antioxidant activity ranging from 111.03 ± 1.03 to 255.92 ± 0.90 mg TE/100 g according to the DPPH assay [28]. In terms of the DPPH assay, NZYC has antioxidant activity several times greater than other foods of interest due to their antioxidant activity

### 3.5. Amino Acid and Organic Acid Profiles

Sixteen amino acids and seven organic acids were quantified in yacon concentrate; the results are presented in Table 6. The most abundant amino acids present in yacon concentrate were L-arginine, L-glutamic acid, L-proline, L-aspartic acid, and asparagine. Significant differences existed between the three batches of yacon concentrate for all amino acids except glycine. In each instance where significant differences existed, the concentration in batch 2 was significantly higher than the other two batches apart from L-tyrosine, where batch 2 was not significantly different to batch 3. Batch 3 only had significantly greater concentration than batch 1 for L-proline, L-tyrosine, and L-valine.

Interestingly, although L-tryptophan is identified in the literature as a major amino acid in yacon concentrate, it was not detected in the present study. In the literature, glutamine, arginine, alanine, threonine, and valine were identified as being present in yacon and yacon concentrate [7,9]; each of these amino acids were quantified in the present study. L-aspartic acid, L-glutamic acid, and L-proline were not identified in the literature but were present in relatively high concentration (>500 mg/100 g in batch 2) in the present study. This could be due to differences in cultivars used in the literature compared to the present study, as well as other factors such as soil or climatic conditions.

Citric acid was by far the most abundant organic acid detected, present at more than ten times the concentration of malic acid, the next most abundant organic acid. Quinic acid and fumaric acid were present at similar concentrations to malic acid; the concentration of succinic acid was lower, while maleic acid and malonic acid were present at very low concentrations or were not detected.

Literature concerning the organic acid profile of yacon concentrate is limited, though formic acid, fumaric acid, citric acid, and malic acid have been identified as being present in yacon concentrate [9]. Formic acid was not quantified in the present study, while fumaric acid, citric acid, and malic acid were all quantified.

## 4. Conclusions

This study provided a comprehensive chemical analysis and glycaemic indexing of New Zealand yacon concentrate. Although yacon has been cultivated for centuries in South America, it has only been introduced relatively recently to the Western world and gained interest as a potential functional food. Yacon concentrate is one novel product that has arisen as a result of growing interest in yacon; however, it has not been well-discussed in the literature. This present study has added significant knowledge to the literature in terms of the chemical composition of yacon concentrate produced from yacon cultivated in New Zealand. The high phytochemical contents in NZYC, including phenolic and flavonoid compounds with proven antioxidant capacities, as well as the low glycaemic index value, could support its classification as a nutraceutical food product for future new diet therapy applications.

## Figures and Tables

**Table 1 foods-12-00074-t001:** Proximate analysis and sugar profile of NZYC (*n* = 9).

Macronutrient	g/100 g NZYC Batch 1	g/100 g NZYC Batch 2	g/100 g NZYC Batch 3
Protein *	4.744 ± 0.650 ^c^	5.675 ± 0.070 ^ab^	6.634 ± 0.158 ^b^
Moisture *	11.943 ± 0.046 ^a^	11.890 ± 0.361 ^a^	9.648 ± 0.320 ^b^
Ash *	2.642 ± 0.130 ^b^	2.981 ± 0.122 ^ab^	3.364 ± 0.410 ^a^
Carbohydrate	80.671 ± 0.664	79.454 ± 0.387	80.354 ± 0.544
Fructose *	35.798 ± 1.871 ^a^	27.783 ± 2.380 ^b^	36.463 ± 2.757 ^a^
Glucose	8.582 ± 0.730	8.334 ± 0.401	8.069 ± 0.546
Sucrose *	10.327 ± 0.475 ^a^	9.166 ± 0.492 ^b^	9.016 ± 0.410 ^b^
1-Kestose (DP3) *	3.299 ± 0.187 ^b^	4.883 ± 0.077 ^a^	2.767 ± 0.046 ^c^
Nystose (DP4) *	5.397 ± 0.424 ^b^	9.552 ± 0.148 ^a^	4.699 ± 0.224 ^b^
DP5 *	4.083 ± 0.291 ^b^	7.843 ± 0.107 ^a^	3.435 ± 0.113 ^c^
DP6 *	3.237 ± 0.308 ^b^	6.636 ± 0.137 ^a^	2.788 ± 0.143 ^b^
DP7 *	2.588 ± 0.250 ^b^	5.963 ± 0.078 ^a^	1.949 ± 0.105 ^c^
DP8 *	2.179 ± 0.319 ^b^	5.260 ± 0.100 ^a^	1.987 ± 0.097 ^b^
DP9 *	1.515 ± 0.258 ^b^	4.247 ± 0.258 ^a^	n.d.
DP10 *	1.201 ± 0.058 ^b^	3.068 ± 0.241 ^a^	n.d.
DP11 *	0.726 ± 0.163 ^b^	2.076 ± 0.358 ^a^	n.d.
DP12	n.d. **	1.517 ± 0.420	n.d.
DP13	n.d.	1.231 ± 0.387	n.d.

* Significant differences existed between batches (*p* < 0.05). Values with different superscript letters are significantly different. ** n.d. not detected.

**Table 2 foods-12-00074-t002:** Mineral profile of NZYC as determined by microwaveplasma atomic emission spectrometry (*n* = 3).

Mineral	mg/100 g NZYC Batch 1
Potassium	658.366 ± 5.927
Phosphorus	93.121 ± 0.500
Calcium	44.225 ± 4.478
Magnesium	36.628 ± 0.907
Iron	23.809 ± 0.199
Sodium	9.035 ± 0.152
Nickel	0.530 ± 0.047
Zinc	0.461 ± 0.023
Copper	0.107 ± 0.006
Lead	0.050 ± 0.003
Titanium	0.001 ± 0.000
Antimony	n.d. **
Boron	n.d.
Cobalt	n.d.

** n.d. not detected.

**Table 3 foods-12-00074-t003:** Phenolic profile of NZYC as determined by liquid chromatography—mass spectrometry (*n* = 9).

Phenolic Compound	mg/100 g NZYC Batch 1	mg/100 g NZYC Batch 2	mg/100 g NZYC Batch 3
Chlorogenic acid *	9.302 ± 1.203 ^a^	10.361 ± 1.398 ^a^	6.187 ± 0.221 ^b^
Caffeic acid	1.479 ± 0.040	1.540 ± 0.150	1.302 ± 0.069
p-Coumaric acid *	0.141 ± 0.006 ^a^	0.137 ± 0.009 ^a^	0.106 ± 0.006 ^b^
Ferulic acid *	0.012 ± 0.002 ^b^	0.018 ± 0.002 ^a^	0.011 ± 0.001 ^b^
Kaempferol *	0.002 ± 0.000 ^b^	0.010 ± 0.001 ^a^	0.001 ± 0.000 ^c^
Isorhamnetin *	0.001 ± 0.000 ^b^	0.002 ± 0.000 ^a^	n.d. **

* Significant differences existed between batches (*p* < 0.01). Values with different superscript letters are significantly different. ** n.d. not detected.

**Table 4 foods-12-00074-t004:** Total phenolic content of NZYC in gallic acid equivalents (GAE) as determined by the Folin-Ciocalteau assay (*n* = 9).

mg GAE/100 g NZYC Batch 1	mg GAE/100 g NZYC Batch 2	mg GAE/100 g NZYC Batch 3
648.95 ± 10.39 ^b^	564.97 ± 9.28 ^c^	785.11 ± 43.14 ^a^

Significant differences existed between batches (*p* = 0.000145). Values with different superscript letters are significantly different.

**Table 5 foods-12-00074-t005:** Antioxidant activity of NZYC measured in Trolox equivalents (TE) (*n* = 9).

Antioxidant Assay	mg TE/100 g NZYC Batch 1	mg TE/100 g NZYC Batch 2	mg TE/100 g NZYC Batch 3
FRAP	1084.14 ± 18.63	1100.17 ± 135.66	1442.54 ± 259.11
CUPRAC	2735.31 ± 235.47	2970.97 ± 90.18	3085.78 ± 637.45
DPPH	1551.38 ± 334.70	1536.08 ± 64.34	1549.04 ± 251.17

**Table 6 foods-12-00074-t006:** Amino acid and organic acid profiles of NZYC (*n* = 9).

Amino Acid/Organic Acid	mg/100 g NZYC Batch 1	mg/100 g NZYC Batch 2	mg/100 g NZYC Batch 3
Asparagine *	204.634 ± 12.724 ^b^	5.675 ± 0.070 ^ab^	6.634 ± 0.158 ^b^
L-Arginine *	468.015 ± 53.710 ^b^	11.890 ± 0.361 ^a^	9.648 ± 0.320 ^b^
Glutamine *	16.443 ± 1.565 ^b^	2.981 ± 0.122 ^ab^	3.364 ± 0.410 ^a^
L-Serine *	92.116 ± 1.905 ^b^	79.454 ± 0.387	80.354 ± 0.544
Ethanolamine *	4.325 ± 0.274 ^b^	16.673 ± 0.828 ^a^	2.196 ± 0.315 ^c^
Glycine	17.796 ± 1.272	30.315 ± 6.988	20.115 ± 7.234
L-Aspartic acid *	329.152 ± 21.144 ^b^	525.230 ± 10.217 ^a^	212.504 ± 6.217 ^c^
L-Glutamic acid *	213.869 ± 7.187 ^b^	586.153 ± 11.943 ^a^	62.252 ± 3.477 ^c^
b-Alanine *	9.302 ± 1.338 ^b^	10.917 ± 0.427 ^a^	7.399 ± 0.132 ^c^
L-Alanine *	191.228 ± 8.570 ^b^	253.718 ± 17.668 ^a^	94.458 ± 3.057 ^c^
γ-Amino-n-butyric acid *	32.695 ± 3.791 ^b^	65.004 ± 1.792 ^a^	8.046 ± 0.754 ^c^
L-Proline *	139.602 ± 5.686 ^c^	539.356 ± 11.472 ^a^	418.518 ± 25.801 ^b^
L-Tyrosine *	10.642 ± 1.047 ^b^	14.879 ± 1.077 ^a^	18.212 ± 2.203 ^a^
L-Valine *	48.867 ± 2.710 ^c^	71.553 ± 0.389 ^a^	60.853 ± 4.240 ^b^
L-Isoleucine *	80.755 ± 1.442 ^b^	123.588 ± 2.421 ^a^	68.812 ± 4.192 ^c^
L-Leucine *	13.550 ± 1.191 ^b^	18.076 ± 1.277 ^a^	12.900 ± 0.485 ^b^
L-Threonine *	51.431 ± 1.844 ^b^	105.134 ± 4.733 ^a^	19.880 ± 1.791 ^c^
Citric acid	2457.447 ± 268.894	2347.894 ± 72.529	2700.594 ± 84.581
Quinic acid *	113.519 ± 14.386 ^b^	89.109 ± 1.103 ^c^	139.749 ± 4.647 ^a^
Malic acid *	178.393 ± 20.137 ^b^	197.381 ± 6.939 ^b^	235.333 ± 6.879 ^a^
Succinic acid *	15.268 ± 2.909 ^b^	22.464 ± 1.050 ^a^	17.845 ± 0.770 ^b^
Fumaric acid *	77.958 ± 13.226 ^b^	145.534 ± 9.374 ^a^	57.639 ± 9.327 ^b^
Maleic acid	1.824 ± 0.521	1.588 ± 0.199	1.981 ± 0.369
Malonic acid *	n.d. **	6.799 ± 0.604 ^a^	n.d.

* Significant differences existed between batches (*p* < 0.01). Values with different superscript letters are significantly different. ** n.d. not detected.

## Data Availability

The data presented in this study are available on request from the corresponding authors.

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
