# Peer review of "A Comprehensive Chemical and Nutritional Analysis of New Zealand Yacon Concentrate"

_foods, 2022, doi:10.3390/foods12010074_

Round 1

Reviewer 1 Report

In general, the paper presents information that may be useful to others dealing with value addition or use of NZYC. For improves, I wish to suggest the following:

Introduction: 

(1) At present, the introduction is too long and less focused on the research question. There is information dump that is remotely relevant (e.g. L34-45, etc). 

(2) What is the research question that this paper is set to answer? Is there paucity of comprehensive data on the NZ cultivar? 

(3) L85 - 92, is there any effect of environment? 

Results and discussion

To improve readability of the par, it may be better to take "Results and  discussion" together. At present, results are provided in sections that do not link to one another, in terms of the observed trends. There should be a link between the approximate analysis and sugar profile, for example, and GI. Analysis of results may be improved. 

Is the data given on dry weight basis?

Is there a reason why minerals were determined on one batch only?

Tables - check that the HSD letters are indicated correctly, e.g., Table 1, all batches have "b", so why are they different? What does it mean when there are no letters?

What's the explanation for the significant differences across the batches? What's your comment in terms of product quality consistency? Are the differences within acceptable coefficient of variation? 

What about anti-nutritional content? 

Why is it important to separate the level of significance? The text does not discuss it. 

Consider making one or two tables of results to reduce the number and improve flow of manuscript. You can combine the related results together.  

Others:

Define UPW the first time used

Do not refer to the methods in L270 too 276

Author Response

In response to Reviewer #1.

In general, the paper presents information that may be useful to others dealing with value addition or use of NZYC. For improves, I wish to suggest the following:

Thank you very much for your constructive feedback, it helps us to improve the quality of our manuscript.

Introduction: 

(1) At present, the introduction is too long and less focused on the research question. There is information dump that is remotely relevant (e.g. L34-45, etc). 

Changes have been made to remove information that is only remotely relevant (L27-36, L40-41, L114-119). One line has been added (L78-79) to define yacon concentrate as definition was removed in L27-36.

(2) What is the research question that this paper is set to answer? Is there paucity of comprehensive data on the NZ cultivar? 

Yes, there is a paucity of comprehensive data on yacon concentrate produced from the ‘New Zealand’ cultivar. L126-127 has been added to emphasize this

(3) L85 - 92, is there any effect of environment? 

According to the literature, the extent to which genetic factors and the environment affects the chemical composition of yacon cultivars is unknown, though factors such as cultivar, plant age, and harvest and post-harvest conditions are stated to affect the FOS content of yacon concentrate as mentioned in the discussion.

Results and discussion

To improve readability of the par, it may be better to take "Results and discussion" together. At present, results are provided in sections that do not link to one another, in terms of the observed trends. There should be a link between the approximate analysis and sugar profile, for example, and GI. Analysis of results may be improved.

We appreciate the Reviewer’s suggestion to combine the results and discussion sections and have done so.                    

Is the data given on dry weight basis?

Data is given on wet weight basis; information has been added to L272 to state this.

Is there a reason why minerals were determined on one batch only?

The mineral content in the New Zealand yacon cultivar is very similar to that of reported literature values from other yacon cultivar from other countries. So, we found that reporting the results from one batch was sufficient.

Tables - check that the HSD letters are indicated correctly, e.g., Table 1, all batches have "b", so why are they different? What does it mean when there are no letters?

Error in Table 1 has been corrected; batch 1 should have had ‘c’ for protein content. If there are no letters, then no significant differences have been identified between the batches for that component, e.g. carbohydrate.

What's the explanation for the significant differences across the batches? What's your comment in terms of product quality consistency? Are the differences within acceptable coefficient of variation? 

Significant differences across the batches could exist for a number of reasons, including time in season of harvest of the yacon roots used to produce each batch of yacon concentrate. Although Yacon New Zealand was approached for information on batch production, the information was not made available. There is evident inconsistency between the three batches which was also observed physically in terms of colour and viscosity.

Generally speaking, differences between the batches were within acceptable coefficients of variation (CoV), with only 8 values having a higher CoV value than 20.00.

  • 121 CoV values (77.56%) fell between 0.00 and 9.99
  • 27 (17.31%) fell between 10.00 and 19.99
  • 6 (3.85%) fell between 20.00 and 29.99 (DP11 batch 1, DP12 batch 2, CUPRAC batch 3, DPPH batch 1, glycine batch 2, maleic acid batch 1)
  • 2 (1.28%) fell between 30.00 and 35.96 (DP13 batch 2, glycine batch 3).

What about anti-nutritional content? 

Anti-nutritional factors were outside the scope of this study

Why is it important to separate the level of significance? The text does not discuss it. 

Separation of level of significance has been removed

Consider making one or two tables of results to reduce the number and improve flow of manuscript. You can combine the related results together.  

Number of tables has been reduced where sections have been combined

Others:

Define UPW the first time used

UPW has been defined as ultrapure water in L177-178.

Do not refer to the methods in L270 too 276

Reference to the methods in this paragraph have been removed

Reviewer 2 Report

The article is interesting and is written very clearly and comprehensibly.

I have the following recommendations for the article:

 - In the introduction, it would be appropriate to describe the method of syrup production in more detail.

- Materials and methods – I recommend specifying the syrup production process in more detail, especially the method of syrup concentration to 80°Brix can affect the final composition.

- A detailed description of Kjeldahl's method is not necessary - the method is very well known and it is better to refer to another article/international norm, etc.

- Why was the profile of mineral substances not also measured in batches 2 and 3?

Author Response

In response to Reviewer #2

The article is interesting and is written very clearly and comprehensibly.

I have the following recommendations for the article:

Thank you very much for your constructive feedback, it helps us to improve the quality of our manuscript.

 - In the introduction, it would be appropriate to describe the method of syrup production in more detail.

We appreciate the Reviewer’s suggestion to describe the method of syrup production in more detail; information from the literature has been added to L78-82.

- Materials and methods – I recommend specifying the syrup production process in more detail, especially the method of syrup concentration to 80°Brix can affect the final composition.

We appreciate Reviewer’s concern regarding the methodology of producing yacon syrup. We have an agreement with the NZ Yacon supplier that they do not want to disclose their methodology for producing the yacon concentrate due to trade secret. However, we can let the reviewer know that freeze drying was involved in the processing

- A detailed description of Kjeldahl's method is not necessary - the method is very well known and it is better to refer to another article/international norm, etc.

Specifics of the Kjeldahl method have been removed.

- Why was the profile of mineral substances not also measured in batches 2 and 3?

The mineral content in the New Zealand yacon cultivar is very similar to that of reported literature values from other yacon cultivar from other countries. So, we found that reporting the results from one batch was sufficient.